# Metabolic Robustness to Growth Temperature of a Cold-Adapted Marine Bacterium

Christopher Riccardi,[a] Marzia Calvanese,[b] Veronica Ghini,[c,d] Tania Alonso-Vásquez,[a] Elena Perrin,[a] Paola Turano,[c] Giorgio Giurato,[e,f] Alessandro Weisz,[e,f] Ermenegilda Parrilli,[b] Maria Luisa Tutino,[b] Marco Fondi[a,g]

aDepartment of Biology, University of Florence, Florence, Italy

bDipartimento di Scienze Chimiche, University of Naples, Napoli, Italy

cCenter of Magnetic Resonance (CERM), University of Florence, Florence, Italy

dConsorzio Interuniversitario Risonanze Magnetiche di Metallo Proteine (CIRMMP), Florence, Italy

eDepartment of Medicine, Surgery and Dentistry 'Scuola Medica Salernitana', University of Salerno, Baronissi, Italy

fGenome Research Center for Health - GRGS, Baronissi, Italy

gCentro interdipartimentale per lo Studio delle Dinamiche Complesse (CSDC), University of Florence, Italy

Christopher Riccardi, Marzia Calvanese, and Veronica Ghini contributed equally to this study. Author order was determined alphabetically based on their first name.

**ABSTRACT**  Microbial communities experience continuous environmental changes, with temperature fluctuations being the most impacting. This is particularly important considering the ongoing global warming but also in the "simpler" context of seasonal variability of sea-surface temperature. Understanding how microorganisms react at the cellular level can improve our understanding of their possible adaptations to a changing environment. In this work, we investigated the mechanisms through which metabolic homeostasis is maintained in a cold-adapted marine bacterium during growth at temperatures that differ widely (15 and 0°C). We have quantified its intracellular and extracellular central metabolomes together with changes occurring at the transcriptomic level in the same growth conditions. This information was then used to contextualize a genome-scale metabolic reconstruction, and to provide a systemic understanding of cellular adaptation to growth at 2 different temperatures. Our findings indicate a strong metabolic robustness at the level of the main central metabolites, counteracted by a relatively deep transcriptomic reprogramming that includes changes in gene expression of hundreds of metabolic genes. We interpret this as a transcriptomic buffering of cellular metabolism, able to produce overlapping metabolic phenotypes, despite the wide temperature gap. Moreover, we show that metabolic adaptation seems to be mostly played at the level of few key intermediates (e.g., phosphoenolpyruvate) and in the cross talk between the main central metabolic pathways. Overall, our findings reveal a complex interplay at gene expression level that contributes to the robustness/resilience of core metabolism, also promoting the leveraging of state-of-the-art multi-disciplinary approaches to fully comprehend molecular adaptations to environmental fluctuations.

**IMPORTANCE**  This manuscript addresses a central and broad interest topic in environmental microbiology, i.e. the effect of growth temperature on microbial cell physiology. We investigated if and how metabolic homeostasis is maintained in a cold-adapted bacterium during growth at temperatures that differ widely and that match measured changes on the field. Our integrative approach revealed an extraordinary robustness of the central metabolome to growth temperature. However, this was counteracted by deep changes at the transcriptional level, and especially in the metabolic part of the transcriptome. This conflictual scenario was interpreted as a transcriptomic buffering of cellular metabolism, and was investigated using genome-scale metabolic modeling. Overall, our findings reveal a complex interplay at gene expression level that contributes to the

Address correspondence to Marco Fondi, marco.fondi@unifi.it.

The authors declare no conflict of interest.

robustness/resilience of core metabolism, also promoting the use of state-of-the-art multi-disciplinary approaches to fully comprehend molecular adaptations to environmental fluctuations.

**KEYWORDS** cold-adaptation, genome-scale modeling, metabolomics, transcriptomics

Microorganisms are able to colonize virtually every environmental niche on Earth (1). They have adapted for millions of years prospering under conditions such extreme as water boiling or freezing points, high radiation, acidic or alkaline pH values, heavy metal pollution, and high salinity (2). Growth temperature, in particular, is one of the environmental parameters that mostly impact the physiology of microorganisms and that is thought to have played a key role in their adaptation, selection, and diversification (3). Given the geological history of our planet, it is reasonable to think that adaptation to changing temperatures has independently occurred many times in evolution; consequently, there exists a vast array of molecular strategies for this purpose, disseminated in the microbial kingdom (4–6). Their characterization is key in this phase of Earth's life, as global change is imposing rapid/drastic modifications in basic environmental parameters (including temperature) that, in turn, will solicit the activation of such temperature-adaptation related pathways in microbial communities. Understanding which genes get activated or which compounds get secreted in the environment following an increase of water temperature will help us model and predict the scenarios of microbial communities in a changing environment.

Without necessarily invoking global change, the need to rewire cellular networks in response to temperature shifts is likely a common feature in natural microbial communities. The temperature in the Southern Ocean, for example, is anywhere from -2 to 10°C, since Antarctic water temperature fluctuation responds to the seasonal advance and retreat of sea ice (7). Therefore, marine microorganisms are exposed to seasonal oscillations in temperature. Upper-ocean microbes can experience a higher variability in sea-surface temperature and their working temperatures exceed the *in situ* Eulerian temperature range by up to 10°C (8). Recent findings demonstrate how even upper-ocean microbes experience along-trajectory temperature variability up to 10°C greater than seasonal fluctuations as a result of large-scale climate variability, indicating a remarkable thermal tolerance by the drifting microbial populations in fluctuating marine environments (8, 9). From a cellular viewpoint, previous studies of cold- and heat-adapted microbes have revealed a variety of molecular adaptations that allow their activity and survival under extreme conditions. These initially involve the change in the expression of specific gene sets but, ultimately, the regulatory changes imposed by temperature increase/decrease are mostly implemented at the metabolic level, since this represents, in the words of Prof. Oliver Fiehn, "the ultimate response of biological systems to genetic or environmental changes" (10). These variegate adaptation strategies observed in temperature-stressed microorganisms indicate that: (i) each microorganism may follow a peculiar route to maintain cellular homeostasis when facing temperature fluctuations (11–13), and (ii) there exists intense cross talk between regulatory and metabolic networks in the response thereof (14).

Here, we specifically investigate the molecular mechanisms through which metabolic homeostasis is maintained in a marine bacterium that (in its natural settings) is known to experience such broad seasonal temperature fluctuations. Indeed, rather than studying the consequences of a cold shock event (a circumstance that rarely occurs in nature), we here focus on the comparison of the main cellular networks in cells growing at 2 different temperatures. This was performed through the integration of metabolomic and transcriptomic data unitedly with genome-scale metabolic modeling of the Antarctic bacterium *Pseudoalteromonas haloplanktis* TAC125 (*Ph*TAC125). This bacterium (15) has been isolated from an Antarctic coastal seawater sample collected in the vicinity of the French Antarctic station Dumont d'Urville, Terre Adélie (66° 40' S; 140° 01' E) and has received much attention in the last decade due to the interest

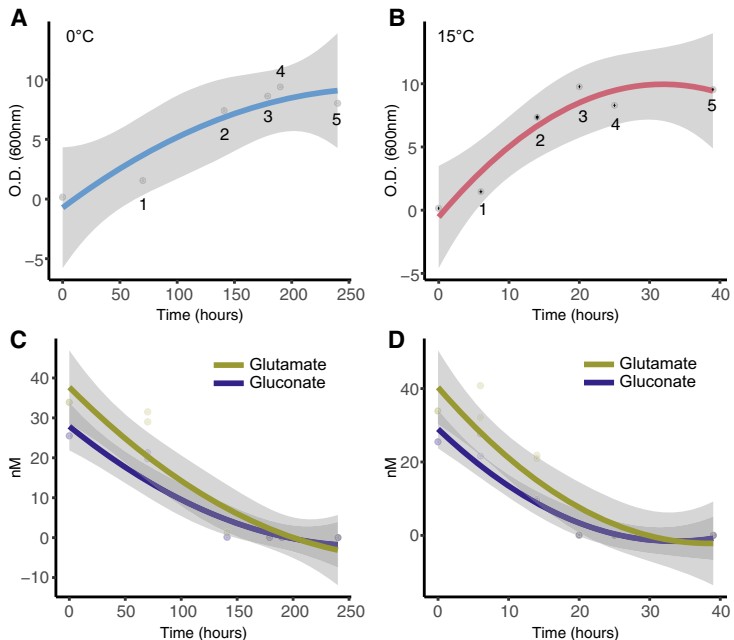

**FIG 1** (A) Growth curve of *Ph*TAC125 at 0°C. Numbers indicate the sampling points for metabolomic and transcriptomic experiments. (B) Growth curve of *Ph*TAC125 at 15°C. Numbers indicate the sampling points for the metabolomic experiments. The transcriptome of *Ph*TAC125 growing cells was sampled at time point 1. (C) Glutamate and gluconate uptake at 0°C. (D) Glutamate and gluconate uptake at 15°C.

in characterizing its cold-adaptation and nutritional adaptation strategies, as well as its biotechnological potential (16–19).

We show that different growth temperatures induce broad transcriptional changes that involve genes of many key metabolic pathways. This transcriptional rewiring, however, is scarcely reflected at the level of the core metabolism, as most key central metabolites show overlapping trends at the 2 tested temperatures. The obtained -omics data were used to compute the flux distributions sustaining growth at low and high temperature, and this provided a mechanistic understanding of the possible adaptation strategies to temperature fluctuations.

## RESULTS

We cultivated *Ph*TAC125 cells in a bioreactor and sampled the 0 and 15°C growth curves at 5 different time points (Fig. 1A and B) that overall resembled the same physiological conditions for the 2 experiments. For the 0°C growth, we sampled the following time points: 70, 141, 179, 190, and 240 h. The 15°C curve was sampled at 6, 14, 20, 25, and 39 h. The average growth rate of the 2 cultures varied, being 0.11 (standard deviation (s.d.) 0.011) h$^{-1}$ and 0.016 (s.d. 6.4e$^{-05}$) h$^{-1}$ at 15° and 0°C, respectively. The same calculation was repeated and limited to the exponential phases of the two cultures, yielding 0.27 (s.d. 0.006) h$^{-1}$ and 0.027 (s.d. 0.0009) h$^{-1}$ at 15° and 0°C, respectively. The samples obtained were used for intracellular and extracellular metabolites quantification through nuclear magnetic resonance (NMR) (see Materials and Methods). For 2 of these time points (Fig. 1A and B, labeled as "1"), we also performed quantified gene expression levels using RNA-Seq.

**PhTAC125 metabolome is qualitatively and quantitatively robust to temperature shift.** Overall, we assigned and analyzed the concentration of 34 intracellular metabolites in the 2 growth curves. These metabolites represented key intermediates of central metabolic pathways, such as the TCA cycle, amino acids biosynthesis, glycolysis, Pentose Phosphate Pathway (PPP), and nucleic acids biosynthesis. Our NMR metabolomic approach was not able to discriminate between the oxidized and reduced forms of NAD and NADP, so we generally refer to NADX and NADPX in the rest of the manuscript. First, we monitored the overall trends of their intracellular concentrations with

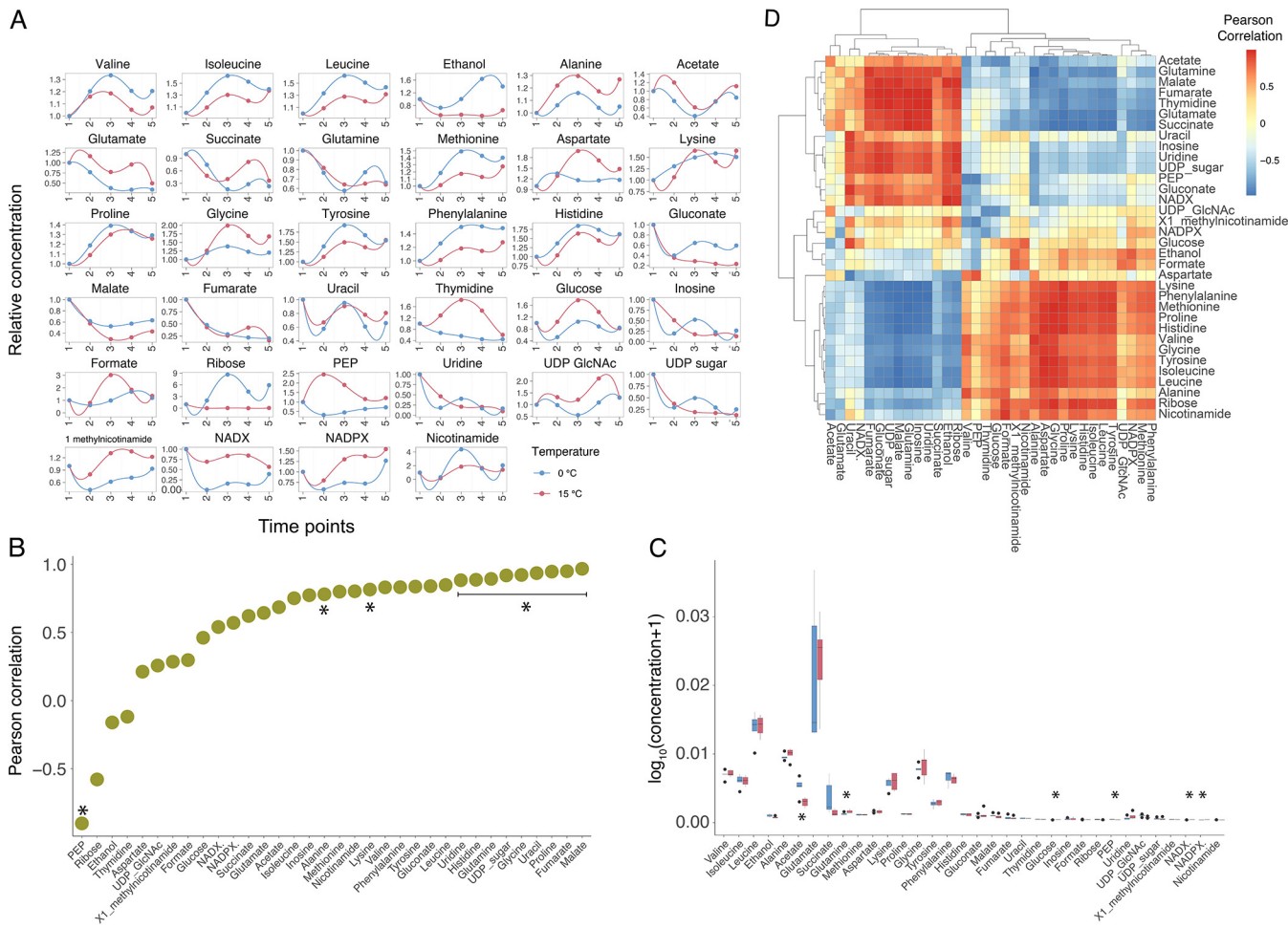

**FIG 2** (A) Normalized intracellular metabolites concentration across the 5 time points. (B) Correlation of each intracellular metabolite at 0 and 15°C (asterisks indicate Spearman correlation *P* value < 0.05). (C) Comparison of the concentration of each intracellular metabolite at 0 and 15°C (asterisks indicate statistically significant correlations, i.e., *P* value < 0.05). (D) All-against-all correlations between intracellular metabolites at 0 and 15°C.

respect to the beginning of the growth experiment (Fig. 2A) by computing the Pearson product moment for each metabolite across the five time points and the relative statistical support (Spearman correlation, *P* value < 0.05 (Fig. 2B)). Strikingly, for about 90% of the analyzed metabolites, we found a positive correlation between the variation of their concentrations during the 2 separate growth experiments (at 0 and 15°C). These positive values ranged from 0.21 in the case of aspartate to 0.96 in the case of malate. Ethanol and thymidine displayed a Pearson product moment correlation (PPM) quite close to zero (-0.16 and -0.1, respectively), thus showing no correlation in the two experiments. Finally, the concentrations of PEP and ribose were negatively correlated in the 2 growth experiments, with PPM of -0.9 and -0.57, respectively. The negative correlation of PEP was supported statistically (Spearman correlation, *P* value = 0.03692).

To qualitatively assess the change in concentration of each metabolite against each other, we computed the all-against-all correlation among the metabolites identified in our study. The results of this analysis are reported in Fig. 2D. We observed 2 main blocks of metabolites: 1 embedding amino acids (with the exception of glutamate and glutamine), and 1 mostly including intermediates of the main central metabolic pathways (TCA, PPP, nucleic acids biosynthesis). Each of these clusters was characterized by a strong (and statistically supported) correlation among the representatives of the same cluster, and an equally relevant anti-correlation with the members of the other cluster. Overall, the intracellular concentration of amino acids was shown to increase over time, whereas the level of, for example, TCA and PPP intermediates was shown to

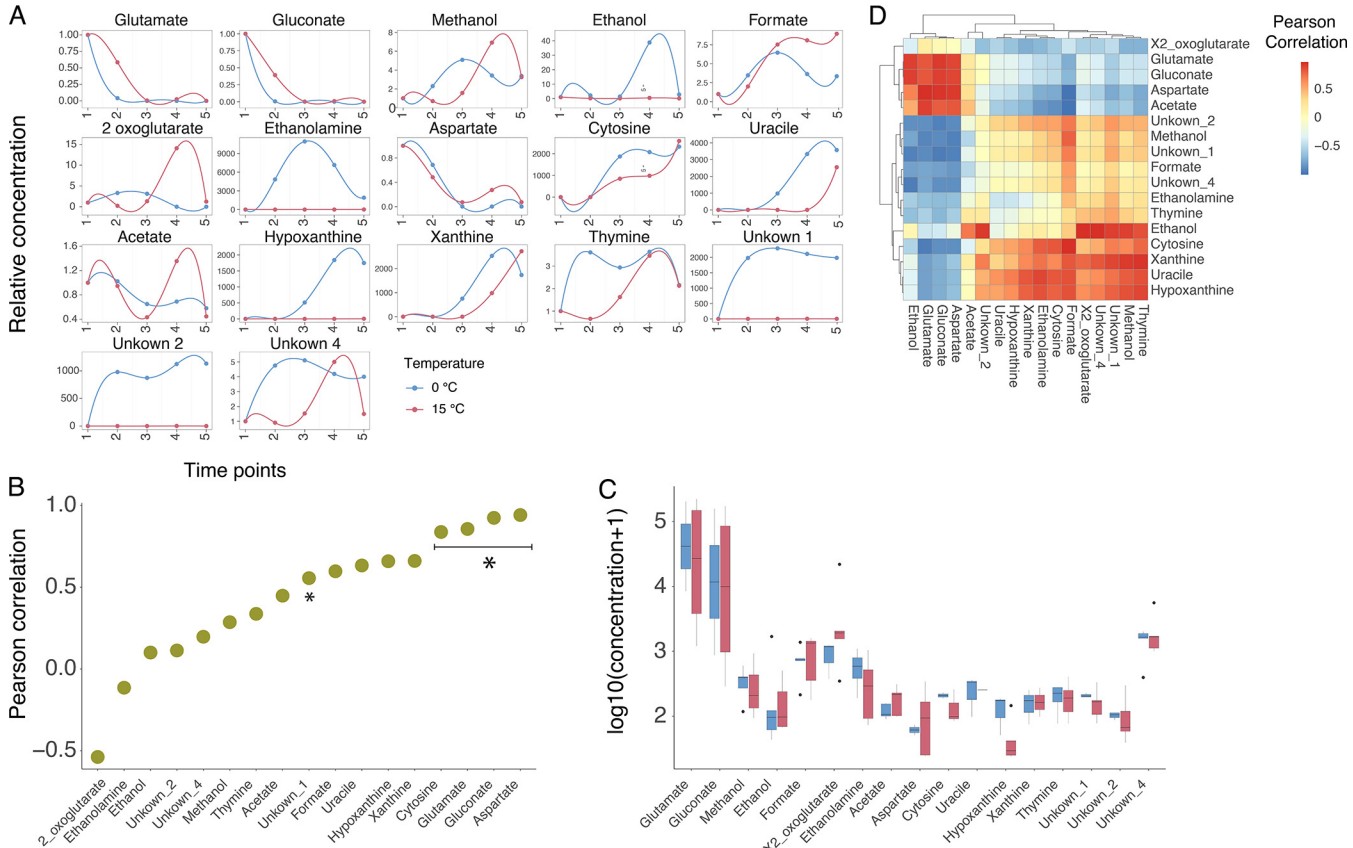

**FIG 3** (A) Normalized extracellular metabolites concentration across the 5 time points. (B) Correlation of each intracellular metabolite at 0 and 15°C (asterisks indicate Spearman correlation $P$ value $< 0.05$) (C) Comparison of the concentration (in a.u.) of each extracellular metabolite at 0 and 15°C (asterisks indicate statistically significant correlations, i.e., $P$ value $< 0.05$). (D) All-against-all correlations between extracellular metabolites at 0 and 15°C.

decrease throughout the growth experiment. To assess whether the growth temperatures had a role in determining the observed trends in metabolite concentrations, we computed an all-against-all correlation for each metabolite at 0° and 15°C separately, and then evaluated whether these 2 correlation matrices differed significantly. The t-tests performed supported no significant differences between the means of these 2 comparisons ($P$ values = 0.74 and 0.89, respectively) suggesting that, despite the large difference in growth temperatures, the overall dynamics of intracellular core metabolites were maintained, pointing to an apparent structural metabolic robustness to growth temperature. This robustness was also conserved quantitatively, as the average concentration of each metabolite was maintained similar across the 2 different growth experiments (Fig. 2C). Indeed, except for 6 metabolites out of 34 (acetate, glutamine, glucose, PEP, NADX, and NADPX, $t$ test, $P$ value $< 0.05$, but $P$ value $> 0.05$ after correction for multiple testing with the Bonferroni method), no other average metabolite concentration showed a significant difference at 0 and 15°C (Fig. 2C and Fig. S1).

A similar scenario was observed for the pool of (17) quantified extracellular metabolites (Fig. 3A). Consistently with growth conditions, glutamate and gluconate concentrations decreased over time until their exhaustion from the growth medium (Fig. 3A), a trend that (i) indicated the almost simultaneous consumption of these 2 nutrients and (ii) was observed at both 0 and 15°C. Out of the 17 extracellular metabolites that were quantified in our experiments, 16 showed a positive correlation between their trends at 0 and 15°C (Fig. 3B). In particular, 10 of them displayed a PPM above 0.5. The only metabolite whose trend differed in the 2 experiments was 2-oxoglutarate (PPM = −0.4). This situation was mirrored at the quantitative level (Fig. 3C). Indeed, none of the extracellular metabolites showed a statistically significant difference between 0°

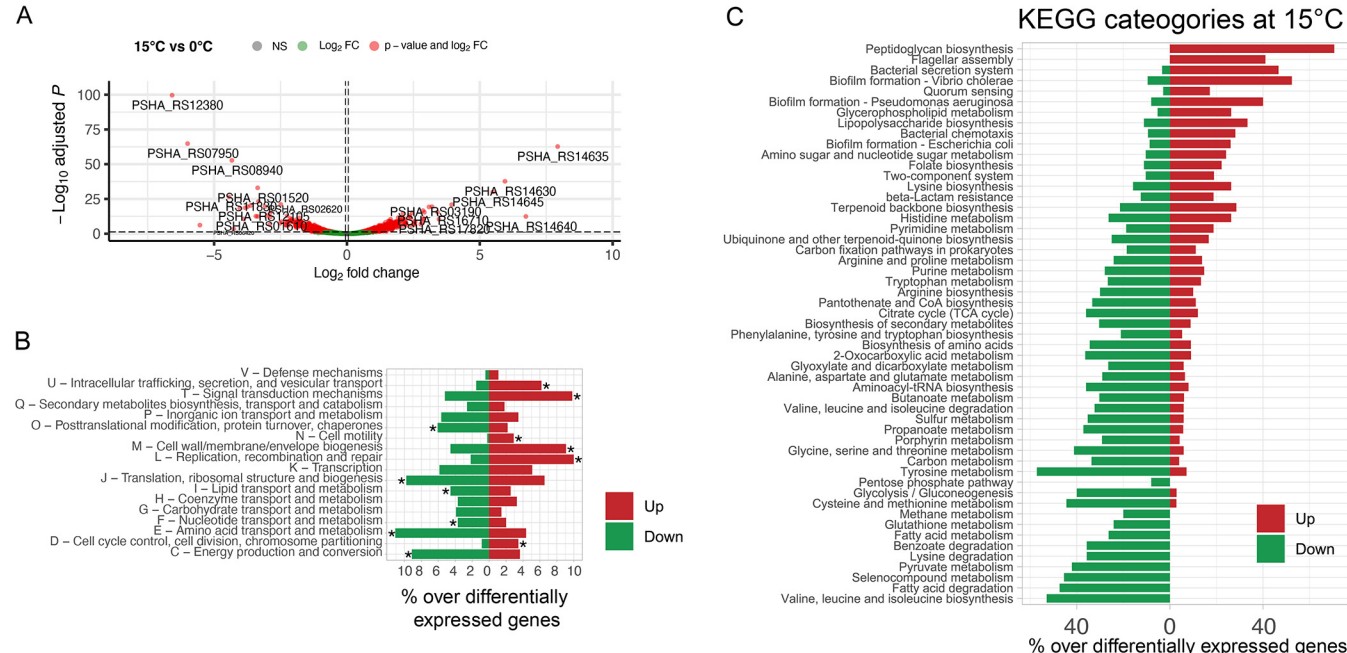

**FIG 4** (A) Volcano plot of up- and downregulated genes. (B) COG categories of up- and downregulated genes (asterisks indicate significantly enriched functional categories). (C) Percentage of differentially expressed genes over the total gene of a subset of *Ph*TAC125 central metabolic pathways. Differentially expressed genes are represented in green (downregulated) or red (upregulated).

and 15°C growth experiments. Finally, metabolomic data allowed us to evaluate the uptake kinetics of glutamate and gluconate in the tested conditions (Fig. 1C and D), revealing that the 2 carbon sources were taken up almost simultaneously in the 2 experiments, and were depleted/exhausted around T4.

**The 0 and 15°C transcriptomes of PhTAC125.** We next characterized the transcriptomes of *Ph*TAC125 at 0 and 15°C. We sampled both curves during exponential growth (Fig. 1A and B, sample points "1"), and sequenced the transcriptome using RNA-Seq technology (see Materials and Methods section). The main features of transcriptomic data are shown in Table S1 and Fig. S2 and S3.

Differentially expressed genes (DEGs) were identified using a $\log_2$ fold change of 1 (or -1) and an adjusted *P* value of 0.05 as thresholds. Overall, we identified 607 differentially expressed genes in the comparison between growth at 15° and 0° C, with an almost identical amount of up- and downregulated genes at 15°C (304 and 303 genes, respectively). The outliers of up- and downregulated genes are shown in Fig. 4A and described in Table 1. Remarkably, out of 607 DEGs, 359 were metabolic protein-coding genes (roughly 59%).

Here we first describe specific features of outlier DEGs, and then focus our attention on broader functional categories that embedded the highest fraction of differentially expressed genes. Among the top 5 downregulated genes, 2 lacked a clear functional annotation (PSHA_RS12380 and PSHA_RS08940), and further experiments are required to understand their role in the adaptation to growth at warmer temperatures. Two of them (PSHA_RS07950 and PSHA_RS06965) were annotated as TonB-dependent receptors, suggesting their involvement in the uptake and transport of large substrates, possibly siderophores complexes and/or vitamins. The remaining gene (PSHA_RS11830) is annotated as NAD-dependent succinate-semialdehyde dehydrogenase, responsible for the conversion of succinate-semialdehyde to succinate. The downregulation of this gene is in line with the lower intracellular concentration of succinate at 15°C, as discussed later in the text. Strikingly, among the over-expressed genes during growth at 15°C, we found a prevalence of cold shock related proteins (4 out of 5) and a RNase R encoding gene (PSHA_RS14645). Although no functional studies have been done on cold shock responses and cold shock proteins (CSPs) from psychrophilic bacteria,

**TABLE 1** The list of the top-five down- and upregulated genes and their functions

| Locus tag | log$_2$FC | $P_{adj}$ value | Annotation |
|---|---|---|---|
| PSHA_RS12380 | −6.58 | 2.43e-100 | Hypothetical protein |
| PSHA_RS07950 | −5.99 | 1.52e-65 | TonB-dependent receptor |
| PSHA_RS06965 | −5.53 | 6.72e-07 | TonB-dependent receptor |
| PSHA_RS11830 | −4.44 | 2.64e-27 | NAD-dependent succinate-semialdehyde dehydrogenase |
| PSHA_RS08940 | -4.33 | 1.85e-53 | PA2169 family four-helix-bundle |
| PSHA_RS14635 | 7.93 | 1.99e-63 | Cold-shock protein |
| PSHA_RS14640 | 6.73 | 4.02e-13 | Cold-shock protein |
| PSHA_RS14630 | 5.94 | 2.03e-38 | Cold-shock protein |
| PSHA_RS14645 | 5.48 | 9.73e-31 | Ribonuclease R |
| PSHA_RS16715 | 3.94 | 1.25e-21 | Cold-shock protein |

similarities with the CSPs that are produced in mesophiles have been observed. In particular, the over-expressed CSPs in *Ph*TAC125 share an amino acid sequence identity ranging between 63.8% and 65.6% with the mesophilic *cspE* of *E. coli*. Furthermore, these proteins contain highly conserved RNA-binding motifs, RNP1 (K-G-F-G-F-I) and RNP2 (V-F-V-H-F) (20, 21), indicated by a black box in Fig. S4.

PSHA_RS14630, PSHA_RS14635, PSHA_RS14640 and PSHA_RS16715 contain 1 highly conserved nucleic acid-binding domain, called cold shock domain (CSD, http://pfam.xfam.org/family/PF00313), which is annotated as RNA chaperone/anti-terminator. Moreover, the NCBI identifies both genes PSHA_RS14635 and PSHA_RS14640 as encoding for the same protein (WP_011329604.1), while the others encode for proteins with a different identifier but virtually same sequence. Cold shock proteins would normally counteract the deleterious effects of temperature drop, enabling the cells to grow at low temperatures (22). Jiang et al. proposed the role of *cspA* as an RNA chaperone capable of melting the RNA secondary structure (23), thereby enhancing translation of mRNAs at low temperatures. However, counterintuitively with their given name, not all members of the CSP family are cold-inducible, and their expression is activated upon different stresses (24). For this reason, CSPs might be required for bacterial adaptation to environmental changes. The fifth upregulated protein, RNase R, belongs to the RNR family. In *Escherichia coli*, the RNase R consists of a central nuclease domain, 2 cold shock (CSD) domains near the N-terminal region of the protein, an S1 domain and a highly basic C-terminal region (25). Cairrão et al. showed that the *rnr* gene is co-transcribed with flanking genes as an operon induced under cold shock in *E. coli* (locus tag b4179) (26). There is an important analogy here since, although *Ph*TAC125's operon map isn't available, its top 4-upregulated genes are mapped consecutively on the genome, with coding sense on the same strand (+), spaced by an average distance of 216 nucleotides. The homology to a variety of domains involved in stress response, together with our observation of a strong upregulation during growth at 15°C, may indicate that *Ph*TAC125 strives to mediate the elimination of detrimental secondary structures and a temperature rise promotes the expression of enzymes that are required for the correct processing of rRNA precursors.

Extending the analysis to a broader functional level, we identified significant differences in 12 COG categories. Specifically, intracellular trafficking, secretion and vesicular transport, signal transduction mechanisms, cell motility, cell wall/membrane/envelope biogenesis, replication, recombination and repair and cell cycle control, cell division, and chromosome partitioning functional categories were found to be over-represented among upregulated genes during growth at 15°C. This matches the results from recent works on the characterization of the evolutionary pathways responsible for thermal adaptation, and the overall notion that one of the main effects of temperature increases on cell physiology consists in the disruption of membrane integrity caused by increased fluidity (27, 28). Overexpression of cell wall and membrane biogenesis-related genes might help overcome this feature associated with growth at higher temperatures.

Conversely, genes involved in post-translational modification, protein turnover,

chaperones, translation, ribosomal structure and biogenesis, lipids, nucleic acids and amino acids transport and metabolism, as well as energy production functional categories, were over-represented among downregulated genes during growth at 15°C (Fig. 4B). We believe that this analysis depicts the system-level adaptation of bacterial life to diverse growth temperatures, and is in line both with the physiological features observed in this work and with previously obtained data on cold/warm adaptation. Indeed, the significant over-expression of genes broadly related to cell replication (COG categories L, M, and D) is in line with the higher growth rate observed at 15°C in respect to 0°C, over the entire growth period (Fig. 1 and B), and in the specific time interval where RNA was sampled. Further, the over-expression of cell motility at warmer temperatures is in perfect agreement with previous assays on *Ph*TAC125 motility that had shown a reduced swimming capability of this strain at 0°C (18).

Surprisingly, the upregulation of cell growth related processes is counteracted by a general downregulation of cellular metabolism. Key processes of *Ph*TAC125 core metabolism such as amino acid, lipid, and nucleotide metabolism together with energy production/conversion, were found to be downregulated (Fig. 4C). As mentioned previously, among 607 DEGs, 359 (59.14%) were metabolic genes. Zooming in at the level of the single pathways (Fig. 4C) revealed that the downregulation affected most of the key central pathways of *Ph*TAC125 metabolic network (including, for example, glycolysis, TCA cycle, PPP, and amino acids biosynthesis genes). In these pathways, the number of downregulated genes strongly outpaced that of upregulated genes (Fig. 4C). Thus, despite consistent intracellular and extracellular metabolic profiles between the 2 tested growth temperatures, the underlying expression of metabolic genes showed an opposite trend, with a remarkable number of DEGs in the 2 growth conditions. For example, the intracellular pools of key TCA intermediates (fumarate and malate) (Fig. 2C) were shown to be similar between 0 and 15°C, despite more than 50% of the TCA genes being differentially expressed in the 2 conditions (Fig. 4C). Similarly, the intracellular concentration of amino acids was shown to be comparable between the 2 conditions (Fig. 2C), whereas the level of expression of their corresponding biosynthetic genes was shown to be significantly different (Fig. 4C).

Accordingly, we hypothesize that the transcriptional network of *Ph*TAC125 provides a buffering mechanism through which metabolic homeostasis is maintained, at least at the level of the central metabolism. In the next section we will combine metabolomic and transcriptomic data with a genome-scale metabolic reconstruction of *Ph*TAC125 to unravel the intimate mechanisms through which this balance is achieved.

**Genome-scale modeling of growth temperature adaptation.** A genome-scale metabolic reconstruction exists for the strain *Ph*TAC125 (16). We, thus, exploited this resource to get a mechanistic interpretation of metabolic homeostasis. First, we checked whether the model was able to represent the experimentally determined phenotypes. We thus constrained the *Ph*TAC125 genome-scale reconstruction with uptake rates of glutamate and gluconate measured at 0 and 15° degrees, and ran an FBA simulation to predict the cellular growth rates. Average uptake rates of glutamate and gluconate were respectively computed between T1 and T3 for both growth curves (that is after 14 and 141 h for 15 and 0°C, see Materials and Methods), and resulted in 0.10 and 0.08 mmol/gCDW*h$^{-1}$ at 0°C and 0.47 and 0.62 mmol/gCDW*h$^{-1}$ at 15°C.

As shown in Fig. 5A, at 0°C the results of this simulation (0.021h$^{-1}$) were in line with the measured growth rates (0.027h$^{-1}$). Conversely, at 15°C a discrepancy was observed between the experimental growth rate and the one predicted *in silico*, 0.27 versus 0.13 h$^{-1}$, respectively. Although there could be many possible explanations for this, we hypothesize that constraining the model only based on the uptake rates of the provided nutrients may not be enough to correctly represent the metabolic phenotype of the cells exposed to different temperatures. For this reason, and to provide a more realistic picture of how metabolic homeostasis is maintained at the level of the central metabolism, we combined metabolomic and transcriptomic data with the genome-scale modeling of *Ph*TAC125 metabolism. We decided to focus on T1 (beginning of the

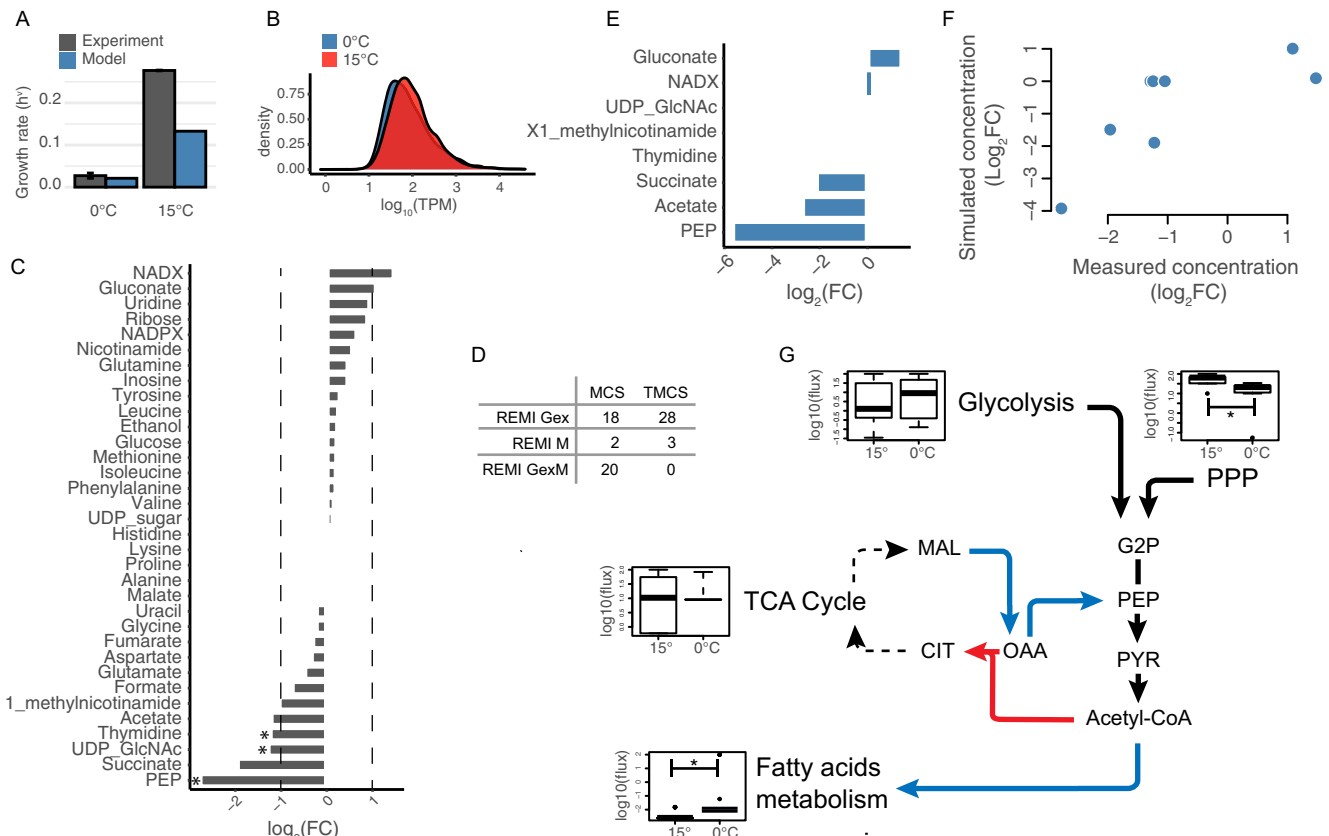

**FIG 5** (A) Comparison between experimental and simulated growth rates. (B) Distribution of TPM values of gene expression at T1. (C) $\log_2$FC of metabolites intracellular concentration at T1 during growth at 15°C versus growth at 0°C. (D) REMI output in terms of Maximal and Theoretical Consistency Scores following transcriptomic and metabolomic data integration. (E) Genome-scale modeling prediction of metabolites $\log_2$FC concentration showing a $|\log_2$FC$| > 1$ during the metabolomic experiment. (F) Correlation between $\log_2$FC of simulated ($y$ axis) versus measured ($x$ axis) internal metabolites concentration. (G) Working model of key metabolic adjustments at the 2 different temperatures. Blue and red arrows indicate fluxes predicted to increase during growth at 0° and 15°C, respectively. Boxplots represent the average flux values for each pathway. Asterisks indicate a significative (i.e., $P$ value $< 0.05$) Kolmogorov-Smirnov statistical test.

exponential growth phase) as this time point should better resemble the cellular physiological state in which FBA assumptions hold the most (i.e., metabolic steady state). Then, we computed the $\log_2$ fold change ($\log_2$FC) of central intracellular metabolites in the 2 conditions (Fig. 5C). Most (76%) of the metabolites showed a $|\log_2$FC$|$ lower than 1, confirming an overall robustness of the central metabolism to growth temperature. Only 8 metabolites displayed a $|\log_2$FC$|$ lower/greater than 1 in the contrast between 15° and 0°C metabolomic data, and for only 3 of them we obtained a statistical support ($P$ value $< 0.05$) (Fig. 5C). More specifically, NADX and gluconate were the 2 metabolites showing a $\log_2$FC $> 1$ (thus being more abundant at 15°) whereas PEP, succinate, UDP-GlcNAc, thymidine, acetate and X1- methyl-nicotinamide displayed a log2FC $< -1$ (thus being more abundant at 0°).

We then constrained the model using all available experimental data obtained in this work using REMI (29) (see Materials and Methods). Briefly, this approach allows translating gene expression and metabolite abundance data resulting from a "perturbation" experiment into differential flux distributions among the 2 resulting conditions. In our case, we used data (metabolomes and transcriptomes sampled at the same time point) from 15° and 0°C growths to analyze the systemic metabolic alteration(s) occurring in this pair of conditions. FC and TPM values (Fig. 5B and C) were used to constrain the model with metabolic and expression data, respectively. Further, the model was also constrained by setting the boundaries of uptake reactions to represent the actual medium used for the experiment (Schatz medium with glutamate and gluconate as the sole carbon sources, see Materials and Methods). The outcome of data integration

into the metabolic reconstruction is summarized by REMI through the computation of the theoretical maximum consistency score (TMCS) and the maximum consistency score (MCS). The first indicates the number of available omics data (for metabolites and reactions), whereas the latter represents the number of those constraints that are consistent with fluxes, and could be integrated into REMI models. As a result, the MCS is always equal to or smaller than the TMCS. In other words, MCS is the largest fraction of available data (metabolomics and transcriptomics) that could be incorporated into an FBA model from a given set of constraints (the abundance of metabolites and transcripts), while ensuring that the model still achieves the required metabolic functionalities and remains feasible. TMCS indicates the number of genes and metabolites with available relative abundance values that can potentially (either because above the specified threshold or their actual inclusion in the metabolic reconstruction) be integrated into the model. The proportion between TMCS and MCS obtained in this work is comparable to that from other studies where REMI was used for the same purposes (29) (Fig. 5D). Out of data integration and FBA simulations, we obtained 2 distinct flux distributions, i.e., the flux distribution resembling growth at 0°C and the 1 theoretically accounting for the growth at 15°C. The differences between the 2 will represent the most likely metabolic alterations in response to growth temperature and, consequently, will highlight those pathways/reactions that contribute the most to maintaining the observed metabolic homeostasis. First, we checked whether our simulations were accounting for the actual differences between 0° and 15°C intracellular metabolomes. We, thus, computed a matching coefficient (Simple Matching Coefficient (SMC)) between the predicted and measured $\log_2$FC of each of the 34 internal metabolites, accounting for how many times the model correctly predicted the increase (or decrease) of its internal concentration. Overall, we found an SMC of 71% between simulated and measured metabolic data, revealing that the model is capable of accounting for most of the central metabolome rewiring in respect to growth temperature. Then, we focused on those metabolites that showed a marked change between growth at 15 and 0°C (i.e., for which $|\log_2$FC$| > 1$) (Fig. 5C). Figure 5F shows that, except for 3 metabolites (UDP-GlcNAc, Thymidine, and 1- methylnicotinamide) for which the model does not predict any difference between the 2 growth conditions, for the remaining 5 metabolites the model correctly predicts a higher (NADX and gluconate) or lower (acetate, PEP and succinate) internal production at 15°. Overall, we found a significant, positive correlation of 0.72 between the measured and the predicted FC of internal concentrations of the 8 metabolites for which $|\log_2$FC$| > 1$ (Fig. 5F) (Spearman correlation, $P$ value = 0.04).

Next, we examined to what extent the central, interconnected pathways showed significantly altered flux distributions. While no clear signal could be identified for glycolysis and TCA cycle (mirroring what we observed with DEGs analysis), both PPP and fatty acids metabolism showed significant differences in their fluxes between 0° and 15° growth simulations. As for fatty acids metabolism, fluxes representative of growth at 0°C were significantly higher (Kolmogorov-Smirnov test, $P$ value = $8.96^{-10}$) than those resembling metabolism at 15°C (Fig. 5F). The opposite was observed for PPP simulated fluxes which displayed, on average, significantly lower values at 0° vs 15°C (Kolmogorov-Smirnov test, $P$ value = 0.0030). A sustained activity of PPP at 15° is in line with the increased internal concentration of gluconate and ribose (2 key PPP intermediates) at this temperature (Fig. 5C). Similarly, an increase in fatty acids biosynthesis (and fatty acids metabolism in general) is in line with previous simulations and experiments concerning the involvement of this process in the adaptation to growth at (relatively) low temperatures (16).

We then focused on the analysis of simulated metabolic fluxes around the metabolite that showed the highest degree of variation, PEP. We asked which metabolic rewiring could lead to an increased production of this metabolite at 0°C. The analysis of fluxes revealed an increased activity at 0°C with respect to 15°C of both the reaction converting L-malate to 2-oxaloacetate ([S]-malate:NAD+ oxidoreductase) and the reaction converting 2-oxaloacetate to PEP (phosphate:oxaloacetate carboxy-lyase). Also,

ATP:pyruvate, $H_2O$ phosphotransferase reaction displayed an increase in flux at 0°. Overall, this would allow the partial redirection of key TCA cycle intermediates to the production of PEP. At the same time, we also recorded an increased flux at 0°C in the reaction redirecting acetyl-CoA to fatty acids biosynthesis. Conversely, reactions leading to the production of citrate from OAA and Acetyl-CoA resulted to be less active at 0°C.

Overall, our experimentally constrained simulations seem to suggest a working model for the metabolic adaptation to growth at different temperatures. Considering growth at 0°C, an increased level of intracellular PEP observed experimentally might be functional to its conversion to acetyl-CoA and its consequent tunneling into fatty acids metabolism (Fig. 5G), and might be the outcome of fluxes redirection from the TCA cycle to PEP production, rather than from glycolysis or PPP. Consistent with this idea is the overall increase of key TCA intermediates (Fig. 5A) measured during growth at 0°C with respect to growth at 15°C that suggests an overall increased activity of this central pathway in the 0°C growth condition.

## DISCUSSION

In this work, we have studied how a cold-adapted bacterium rewires its central metabolism when growing at 2 distinct temperatures that overall resemble a seasonal shift. This was done by characterizing a pool of 34 intracellular and 17 extracellular central metabolites during 5 different time points of its growth curves (i.e., at 0°C and 15°C) and by evaluating gene expression during the initial stages of its exponential phases. To our surprise, the top 5 upregulated genes at 15°C contain a cold shock domain, which is usually recruited to counteract the deleterious effects of temperature drop. A possible explanation for the apparent paradox where CspA is activated, not only following cold stress, but also under non-stress and other stress conditions which entail a downregulation of bulk gene expression and protein synthesis is presented in (30). CSPs have been found in almost all types of bacteria and are mainly induced after a rapid temperature downshift to regulate the adaptation to cold stress but are also present under normal conditions to regulate other biological functions (31). For example, in *E. coli*, only 4 (*cspA*, *cspB*, *cspG*, and *cspI*) of the 9 CSP genes are cold-induced (32). Two of them, *cspE* and *cspC*, are constitutively expressed at physiological temperatures, and act as 'housekeeping RNA chaperones' to modulate the global gene expression (33, 34). Furthermore, these CSPs are also involved in the transcription antitermination mechanism, which is based upon preventing the formation of secondary structures on the nascent mRNA (35–37). In *Ph*TAC125, four cold shock-like proteins *cspE* are upregulated during growth at 15°C. This indicates that these CSP proteins may not play a role in cold-adaptation in *Ph*TAC125 but, analogously to *E. coli*, they may act as chaperones by destabilizing secondary structures in target RNA at high temperature so that the single-stranded state of target RNA is maintained. This may then enable efficient transcription and translation.

From a broader perspective, the comparison between the 0° and 15°C transcriptomes produced more than 600 differentially expressed genes, assigned to a dozen of different functional categories. On the other hand, the intracellular concentration of nearly 90% of the analyzed metabolites correlates positively in the 2 conditions. We interpret this as the capability of *Ph*TAC125 regulatory network to buffer the temperature shift to produce strikingly similar metabolic phenotypes, despite the temperature gap. This is also reflected in the conservation of the overall metabolic network structure (i.e., the presence of correlated clusters of metabolites) (Fig. 2C and Fig. 3C). Remarkably, this conservation involved metabolites from many different pathways, such as amino acids metabolism, TCA cycle, PPP, and glycolysis, suggesting that such robustness is propagated at the level of the entire metabolism and is not restricted to specific pathways. Studies on the metabolic response to growth at different temperatures in other microorganisms have generally shown larger variability among intracellular pools of metabolites (38), and/or the tendency to activate genetic mechanisms that shut down metabolism under longer-term high temperature stress (39). This is not the case for

PhTAC125 that, through a global transcriptional buffering, maintains very consistent trends of its intracellular and extracellular metabolomes. This mirrors, for example, the response of *E. coli* to genetic and environmental perturbations and its capability to maintain metabolite levels stable, reflecting the rerouting of fluxes in the metabolic network (40). A few metabolites stood out in this conservation of metabolite pools, including PEP and ribose (showing opposite trends at 0 and 15°C), and ethanol and thymidine (showing no correlation between the 2 experiments). Also, NADX and NADPX levels differed in the 2 conditions, as shown in Fig. 2A.

To unravel this complex interplay and produce a mechanistic interpretation of this adaptation, -omics data (together with data on carbon sources uptake rates) were used to constrain a genome-scale metabolic reconstruction and derive the most-likely metabolic phenotypes at these 2 temperatures. This approach provided evidence that most of the metabolic adaptation is probably played at the level of the phosphoenol-pyruvate (PEP)–pyruvate–oxaloacetate node and specifically involves the increase of TCA fluxes and their redirection to PEP production. Consistent with this idea, is the overall reduction of key TCA intermediates at 15°C (Fig. 5A) and the observation that, upon growth on non-glycolytic compounds (e.g., glutamate and gluconate) the cycle intermediates malate or oxaloacetate must be converted to pyruvate and PEP for the synthesis of glycolytic intermediates. In our case, however, the availability of an increased pool of PEP would rather allow its conversion to acetyl-CoA, and its consequent tunneling into fatty acids metabolism (Fig. 5G). Also the overexpression of genes involved in valine, leucine, and isoleucine degradation (Fig. 4C) at 0°C can be associated with the higher activity of fatty acids metabolism, as the degradation of these 2 amino acids leads to the production of acetyl-CoA (41). Consistently with this observation, our model predicts an increased activity of fatty acids metabolism at 0° (Fig. 5G). The importance of fatty acids metabolism in growth at low temperatures is largely known. For example, the fluidity of the rigidified membrane (42) imposed by low temperatures can be restored through the modulation of the (i) saturated and unsaturated fatty acids, (ii) fatty acid chain length, and (iii) the proportion of *cis* to *trans* fatty acids thus, ultimately, through the modulation of fatty acids metabolism (as observed in this work). This also validates and extends previous findings on general cold-adaptation strategies (16, 43, 44), pinpointing the role of specific metabolic reactions in this process.

We also found an increased PPP activity at 15° with respect to 0°C (Fig. 5G). Increased PPP activity signifies increased NADPH production, and is linked with many biosynthetic processes. Interestingly, high activity of the PPP pathway has been described as an adaptative mechanism (in soil bacteria) to temperature stress (45), in particular by linking the requirement of sugar units for biofilm formation to the overall activity in this metabolic pathway. We know for a fact from the work of (18) that PhTAC125 displays strong propensity to form more biofilm at 15°C versus 0°C when grown in the same minimal medium used in this work. Also, we found a remarkable fraction of overexpressed biofilm-related genes (Fig. 4C), reinforcing the idea of a temperature-mediated switch to this kind of lifestyle. Thus, we infer that a similar mechanism is at play in this marine bacterium where the redirection of fluxes in PPP at 15°C would be the metabolic basis for the formation of cellular aggregates at higher temperatures. Moreover, the increased PPP activity can respond to other cell requirements linked to the DNA replication recombination and repair, since the ribose produced by PPP is necessary for these cellular activities that result more active at 15°C (Table S2), as suggested by transcriptomic analysis (Fig. 5B).

Very recently, Macarena Toll-Riera et al. published (28) a work in which PhTAC125 was evolved at increasingly higher temperatures to study the evolutionary potential of upper thermal tolerance, and characterize the genomic basis of temperature adaptation. Interestingly, the authors found almost no metabolic genes among those that apparently provided an increased fitness at higher temperatures. Indeed, clones that were selected for their improved growth at higher temperatures (up to 30°C) had mainly mutations in proteases-encoding genes or in genes involved in chromosome

copy number reduction, energy production and conversion and cell wall biosynthesis. Thus, apparently, the metabolism of *Ph*TAC125 was not the primary target of mutations that led to an increased fitness in the tested conditions. This, in turn, may indicate the presence of an already optimized and plastic metabolism that allows growth in a broad range of temperatures, in line with the metabolic robustness that we have characterized in this work.

**Conclusions.** The aim of our experiments was to evaluate the effect of growth temperatures on cellular homeostasis, particularly at the metabolic level. We have shown that a cold-adapted marine bacterium expresses 2 very similar metabolic phenotypes in response to 2 widely different temperatures, by adjusting the expression of key enzymes and fine-tuning the intracellular concentration of key intermediates. This illuminated on the possible molecular mechanisms that marine microbes may use to adapt to broad (seasonal) temperature changes. In particular, we showed that specific changes at the level of regulatory circuits can buffer such variations, and maintain the underlying metabolic network robust to temperature fluctuations. In the future, it will be interesting to investigate whether similar mechanisms are at play in natural microbial assemblages.

## MATERIALS AND METHODS

**Strains and growth conditions.** *Ph*TAC125 (46) cells were grown in a 1.5 L GG medium (47) in a Stirred Tank Reactor 3 L fermenter (Applikon) connected to an eZ2 Bio Controller (Applikon) at 2 different temperatures (0°C and 15°C). This medium contains glutamate and gluconate as the only carbon sources. The bioreactor was equipped with the standard pH-, pO2-, level- and temperature sensors for the bioprocess monitoring. For the growths of the *Ph*TAC125 bacterium, the pre-culture was centrifuged (6000 × *g*, 20 min, 4°C); the cells were washed twice with fresh medium, and then used to inoculate the bioreactor with a starting $OD_{600}$ of 0.2, in aerobic conditions (50% and 30% dissolved oxygen at 15°C and 0°C, respectively), in stirring (500 rpm at 15°C and 250 rpm at 0°C). The bacterial culture was carried out at 15°C for 40 h or at 0°C for 240 h. Each culture condition was repeated three times. Cell growth was monitored, measuring the $OD_{600}$ about every 2 h in the experiments at 15°C, and every 8 h at 0°C. Three different measurements were performed at each time point for each biological replicate. For intra- and extracellular metabolites analysis, the samples were taken in triplicate at 5 different time points during the growth in GG medium at 2 different temperatures (70 h, 141 h, 179 h, 190 h, and 240 h at 0°C, and 6.5 h, 14 h, 20 h, 25 h, and 39 h at 15°C). For the analysis of extracellular metabolites, aliquots (1 mL) of cell cultures were harvested during the growth and centrifuged for 15 min at 13,000g at 4°C; the supernatant was recovered, filtered (Filtropur 0.2 $\mu$m, SARSTED AG & Co. KG) and stored at −80°C. The analysis of intracellular metabolites was performed on 60 $OD_{600}$ pellets recovered during the growth by centrifuging for 20 min at 6,000 rpm at 4°C.

**Transcriptomics.** For the RNA-Seq experiment, 1 $OD_{600}$ pellets were recovered during the exponential growth phase (~1/1.5 OD/mL) at 15°C and 0°C by centrifugation (10 min, 13,000g, 4°C). Cell pellets were washed in RNase-free PBS three times and stored at −80°C. Total RNA was isolated from the cells using the Direct-zol RNA Kit (Zymo Research) following the manufacturer's instructions. Contaminating genomic DNA was then removed through treatment with RNase-free DNase I (Roche).

**RNA sequencing.** RNA concentration was measured using Quant-IT RNA assay kit-high sensitivity and a Qubit Fluorometer (Life Technologies), and its quality and integrity assessed with the Agilent 4200 Tapestation System (Agilent Technologies). Indexed libraries were prepared starting from 400 ng of total RNA according to Universal Prokaryotic RNA-Seq Library Prep kit (Tecan). Final libraries were sequenced at a concentration of 1.7 pM/lane on the NextSeq 500 platform (Illumina Inc) in paired mode 2 × 75bp. For each experimental condition, 3 biological replicates were prepared.

**RNA-Seq data analysis.** The 12 samples were assessed for base call quality and adapter content using fastp (48), allowing down to a mean quality threshold of 20 (i.e., probability of incorrect base call of 1 in 100) and minimum read length of 40 nucleotides. A median of 98.9% of the reads passed quality check, indicating that sequencing was carried out pristinely, and our data were biologically reliable. Salmon (49) index was built on *Ph*TAC125's transcriptome free of tRNA and rRNA sequences, providing the entire genomic sequence as background decoy to account for possible underlying DNA contamination. CDS and genomic FASTA were retrieved on the NCBI, using assembly accession GCF_000026085.1. Moreover, CDS and genomic files were concatenated with data from pMEGA plasmid (NZ_MN400773.1) and pMtBL plasmid (NZ_AJ224742.1). The transcriptome quantification step was performed using the – *validateMappings* flag, to ensure a sensitive selective alignment of the sequencing reads. The resulting mapping rates ranged between 56.8% and 66.21% (average of 62.47%), whereas roughly half of the reads represented ribosomal DNA sequences (discarded for downstream analyses). The integration of transcript-level abundance estimates from Salmon with the data analysis pipeline was performed using R package *tximport* (R Core Team, 2022, https://www.R-project.org/, R version 4.0.3). The testing of changes in the overall transcriptional output was handled through the statistical engine in R package DESeq2, with an expected proportion of false positives set at 5%. The volcano scatterplot showing statistical significance versus magnitude of change for 607 differentially expressed genes was produced using the library EnhancedVolcano from the R Bioconductor 3.14 suite.

**Metabolomics.** ¹H NMR-based metabolomic analyses were performed on cell lysates and growth media to monitor the intracellular metabolites, and the uptake and release of the extracellular metabolites, respectively, by measuring their concentration levels in samples collected at different time points during the cell growth.

Medium samples were prepared in 5.00 mm NMR tubes by mixing 60 $\mu$L of a potassium phosphate buffer (1.5 M $K_2HPO_4$, 100% (vol/vol) $^2H_2O$, 10 mM sodium trimethylsilyl [2,2,3,3$-^2H_4$]propionate (TMSP), pH 7.4), and 540 $\mu$L of each growth medium.

Cell lysate samples were prepared in 5.00 mm NMR tubes by mixing 60 $\mu$L of $^2H_2O$ and 540 $\mu$L of samples. All the NMR spectra were recorded using a Bruker 600 MHz spectrometer (Bruker BioSpin) operating at 600.13 MHz proton Larmor frequency, and equipped with a 5 mm PATXI $^1$H-$^{13}$C-$^{15}$N and $^2$H-decoupling probe including a z axis gradient coil, an automatic tuning-matching (ATM) and an automatic and refrigerate sample changer (SampleJet). A BTO 2000 thermocouple served for temperature stabilization at the level of approximately 0.1 K at the sample. Before measurement, samples were kept for 5 min inside the NMR probe head, for temperature equilibration at 300 K.

¹H NMR spectra were acquired with water peak suppression and a standard NOESY pulse sequence using 128 scans, 65536 data points, a spectral width of 12019 Hz, an acquisition time of 2.7 s, a relaxation delay of 4 s, and a mixing time of 0.1 s.

The raw data were multiplied by a 0.3 Hz exponential line broadening before applying Fourier transformation. Transformed spectra were automatically corrected for phase and baseline distortions. All the spectra were then calibrated to the reference signal of TMSP at $\delta$ 0.00 ppm using TopSpin 3.5 (Bruker BioSpin srl).

The metabolites, whose peaks in the spectra were well resolved, were assigned, and their levels analyzed using a dedicated R script developed in-house. In total, 34 and 17 metabolites were identified and quantified in the cell lysate and in the growth medium spectra, respectively. The assignment was performed using an internal ¹H NMR spectral library of pure organic compounds (BBIOREFCODE, Bruker BioSpin), stored reference NMR spectra of metabolites, and spiking experiments. Matching between new NMR data and databases was performed using the Assure NMR software (Bruker BioSpin). The relative concentrations of the various metabolites were calculated by integrating the corresponding signals in defined spectral ranges, using in-house developed R 3.0.2 scripts. Similarly, downstream data analysis was performed using R. Raw data, post-Raw data, and post-processing codes are made available at https://github.com/combogenomics/MetRob015.

Uptake rates of gluconate and glutamate at the 2 different temperatures were computed as the ratio between the average growth rate between T1 and T3 ($\mu$) and the biomass yield ($\lambda$). The former was computed as:

$$\mu = \frac{logOD_{T3} - logOD_{T1}}{T3 - T1}$$

The latter was computed according to the following relationship:

$$\lambda = \frac{Biomass\ \left(\frac{g}{L}\right)}{Consumed\ C\ source\ (mol)}$$

Biomass was obtained from OD values as described in (50), using 0.74 as a scaling factor for the growth at 15° and 0.66 for the growth at 0°C.

**Genome-scale metabolic modeling.** The genome-scale metabolic reconstruction used in this work is the one recently used in (51). All the simulations were run in MATLAB 2019a, using the COBRA toolbox (52) version 2.7.4. Metabolomic and transcriptomic data were integrated using REMI method (29), providing TPM gene expression and relative concentration values for metabolites at time point T1. Both for metabolites and gene expression, we selected the top 3% as upregulated and the bottom 3% as downregulated (REMI default is 5%). Consistently with the original REMI publication, we used the 2-fold change as the cutoff threshold to identify the significant gene expression and metabolite changes. All the other parameters were left as default. The predicted internal concentration of each metabolite was computed using the *computeFluxSplits* function implemented in the COBRA toolbox. This function computes the relative contributions of fluxes to the net production and consumption of a specific set of metabolites included in the model. Statistical tests on flux distributions were calculated using R.

To estimate the difference in the activity of the main central metabolic pathways at 0° and 15°C, we computed the predicted flux of each reaction in these pathways (TCA cycle, glycolysis, PPP, and fatty acids metabolism) at the 2 different temperatures, and averaged this number of reactions by the total number of reactions included in that pathway. From this latter analysis, we excluded those reactions that (i) had a flux equal to zero (were inactive) in both conditions and (ii) had a different sign (i.e., changed direction) at the 2 temperatures.

**Data availability.** The authors confirm that the data supporting the findings of this study are available within the article and/or its supplemental materials. Collection of sequence data produced in this study is available under the BioProject PRJNA886636.

## SUPPLEMENTAL MATERIAL

Supplemental material is available online only.

**FIG S1**, PDF file, 0.2 MB.

**FIG S2**, PDF file, 0.1 MB.
**FIG S3**, PDF file, 0.04 MB.
**FIG S4**, PDF file, 0.1 MB.
**TABLE S1**, PDF file, 0.01 MB.
**TABLE S2**, PDF file, 0.02 MB.

## ACKNOWLEDGMENTS

We thank Dr. Assunta Sellitto for technical assistance for RNA-Seq libraries preparation.

P.T. and V.G. acknowledge the support and the use of resources of Instruct-ERIC, a Landmark ESFRI project, and, specifically, the CERM/CIRMMP Italy Centre.

This study was supported by Regione Campania, Progetto GENOMAeSALUTE (POR CAMPANIA FESR 2014/2020, azione 1.5; CUP: B41C17000080007), by PNRA (Programma Nazionale di Ricerche in Antartide) grant PNRA18_00075 and PNRA18_00335, and by a PRIN-MUR (RESEARCH PROJECTS OF RELEVANT NATIONAL INTEREST– 2020 Call) Project: 20208LLXEJ.

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
