## [Reviewer comments · mSystems]

Metabolic robustness to growth temperature of a cold adapted marine bacterium

Christopher Riccardi, Marzi Calvanese, Veronica Ghini, Tania Alonso-Vásquez, Elena Perrin, Paola Turano, Giorgio Giurato, Alessandro Weisz, Ermenegilda Parrilli, Maria Luisa Tutino, and Marco Fondi

Corresponding Author(s): Marco Fondi, University of Florence

Review Timeline:

Submission Date:	November 12, 2022
Editorial Decision:	January 3, 2023
Revision Received:	January 23, 2023
Accepted:	January 26, 2023

Editor: Rachel Mackelprang

Reviewer(s): The reviewers have opted to remain anonymous.

Transaction Report:

DOI: <https://doi.org/10.1128/msystems.01124-22>

January 3, 2023

Dr. Marco Fondi
University of Florence
Department of Biology
Florence
Italy

Re: mSystems01124-22 (Metabolic robustness to growth temperature of a cold adapted bacterium)

Dear Dr. Marco Fondi:

Thank you for submitting your manuscript to mSystems. We have completed our review and I am pleased to inform you that, in principle, we expect to accept it for publication in mSystems. However, acceptance will not be final until you have adequately addressed the reviewer comments. In addition to the comments from reviewers, you also must make the following adjustments:

- 1) Add a data availability paragraph (including RNA-Seq data) to the end of the methods. <https://journals.asm.org/open-data-policy>
- 2) Add the importance paragraph to the manuscript text file.

Preparing Revision Guidelines

Sincerely,

Rachel Mackelprang

Editor, mSystems

Journals Department
Reviewer comments:

Reviewer #1 (Comments for the Author):

The paper is very interesting. Only minor comments.

The title is not very clear, in particular the meaning of the term robustness.

I suggest modifying the title with something more informative; for example, by adding that the bacterium used in this study is marine.

There is any particular reason why the bacterium is grown in GG medium?

Glutamate can stimulate the synthesis of fluorescent molecules. Is this the case?

What is 2H₂O? I suggest writing it as H₂O₂.

Reviewer #2 (Comments for the Author):

The work by Riccardi et al. investigated and compared the main cellular networks in cells growing at two different temperatures (0 and 15 °C) through the integration of metabolomic and transcriptomic data unitedly with genome-scale metabolic modelling of a cold-adapted Antarctic bacterium. Overall, the experiments were well designed, the multi-omics approaches used sound, and the results well-represented and discussed. I only have a few minor comments.

Line 39 and 758, a cold-adapted marine bacterium.. This clarification (marine bacterium) is necessary when you argue that your results have implications for the adaptation of marine microbes to broad seasonal temperature changes in natural settings.

Line 85-101, the references in this part is too old and should be updated with new ones.

Line 86 Delille et al., Line 132 (Sannino et al., 2017) (Sannino et al., 2017).. In general, the manuscript is well-written, but a careful checking is still required for these minor errors.

Line 116, citation of the original reference reporting isolation of this bacterium is needed.

Line 118, has received much attention in the last decade.. citation of some references is needed to support this statement.

Line 340-341: Did you conduct a p-value adjustment for t-test?

Finally, the authors should be more careful when extrapolating general trends from culture experiments of a single isolate to natural microbial assemblages.

Reviewer #1 (Comments for the Author):

The paper is very interesting. Only minor comments.

The title is not very clear, in particular the meaning of the term robustness.

I suggest modifying the title with something more informative; for example, by adding that the bacterium used in this study is marine.

Reply: We thank the reviewer for the positive evaluation of our work. After careful consideration, we decided to include the word 'marine' as it renders the title much more informative. Concerning the use of “robustness”, we report here a few commonly accepted definitions: “Robustness is a system’s intrinsic ability to maintain functionality under perturbation (also recalled in Libiseller-Egger et al. 2020, DOI: 10.1038/s41540-020-00155-5 about metabolic systems)” and “Metabolic robustness refers to the ability of a metabolic system to buffer changes in its environment”. In our opinion, this term fits with the results obtained in this work and we would opt for the term 'robustness' since it indicates a rooted tendency of the metabolic apparatus of PhTAC125 to thrive at such a temperature fluctuation. We are open, however, to specific suggestions (also from the Editor) on how to modify the title to make it even more informative.

There is any particular reason why the bacterium is grown in GG medium?

Reply: Yes. As shown in several works (DOIs: 10.1007/s00253-016-7942-5, 10.3390/metabo11080491) the GG medium represents a new and optimized medium for the growth (and the production of recombinant proteins) of this bacterium at a broad range of temperatures. Additionally, being a minimal medium, it offers one of the most important conditions for using constraint-based metabolic modelling, i.e. a control over up-taken carbon sources during laboratory conditions.

Glutamate can stimulate the synthesis of fluorescent molecules. Is this the case?

Reply: We haven’t observed the synthesis of fluorescent molecules in the tested conditions.

What is 2H2O? I suggest writing it as H2O2.

Reply: Thanks for pointing this out. A formatting issue has occurred here. 2H2O should have been ²H₂O, indicating the presence of a deuterium. We have carefully revised the manuscript in this context.

Reviewer #2 (Comments for the Author):

The work by Riccardi et al. investigated and compared the main cellular networks in cells growing at two different temperatures (0 and 15 oC) through the integration of metabolomic and transcriptomic data unitedly with genome-scale metabolic modelling of a cold-adapted Antarctic bacterium. Overall, the experiments were well designed, the multi-omics approaches used sound, and the results well-represented and discussed. I only have a few minor comments.

Reply: We are thankful to the reviewer for the positive evaluation of our work.

Line 39 and 758, a cold-adapted marine bacterium.. This clarification (marine bacterium) is necessary when you argue that your results have implications for the adaptation of marine microbes to broad seasonal temperature changes in natural settings.

Reply: We agree, correction performed

Line 85-101, the references in this part is too old and should be updated with new ones.

Reply: We have modified the text in this part and added a more recent references.

Line 86 Delille at al., Line 132 (Sannino et al., 2017) (Sannino et al., 2017).. In general, the manuscript is well-written, but a careful checking is still required for these minor errors.

Reply: Corrections performed. We have carefully, checked the manuscript for minor errors.

Line 116, citation of the original reference reporting isolation of this bacterium is needed.

Reply: We have included the reference dealing with the preliminary isolation and characterization of *Pseudoalteromonas haloplanktis* TAC125 (DOI: 10.1046/j.1432-1327.2000.01299.x).

Line 118, has received much attention in the last decade.. citation of some references is needed to support this statement.

Reply: We have moved here some references supporting our statement and that were erroneously included elsewhere in the previous version of the manuscript.

Line 340-341: Did you conduct a p-value adjustment for t-test?

Reply: No, we didn't. We have now performed the adjustment indicated by the reviewer and included the information in the new version of the manuscript.

Finally, the authors should be more careful when extrapolating general trends from culture experiments of a single isolate to natural microbial assemblages.

Reply: In revising the manuscript we have made it clearer that our considerations apply to the specific microorganism analysed here and that, in the future, it will be interesting to investigate whether similar mechanisms are at play in natural microbial assemblages.

January 26, 2023

Dr. Marco Fondi
University of Florence
Department of Biology
Florence
Italy

Re: mSystems01124-22R1 (Metabolic robustness to growth temperature of a cold adapted marine bacterium)

Dear Dr. Marco Fondi:

Your manuscript has been accepted, and I am forwarding it to the ASM Journals Department for publication. For your reference, ASM Journals' address is given below. Before it can be scheduled for publication, your manuscript will be checked by the mSystems production staff to make sure that all elements meet the technical requirements for publication. They will contact you if anything needs to be revised before copyediting and production can begin. Otherwise, you will be notified when your proofs are ready to be viewed.

If you would like to submit a potential Featured Image, please email a file and a short legend to msystems@asmusa.org. Please note that we can only consider images that (i) the authors created or own and (ii) have not been previously published. By submitting, you agree that the image can be used under the same terms as the published article. File requirements: square dimensions (4" x 4"), 300 dpi resolution, RGB colorspace, TIF file format.

We recognize that the video files can become quite large, and so to avoid quality loss ASM suggests sending the video file via <https://www.wetransfer.com/>. When you have a final version of the video and the still ready to share, please send it to mSystems staff at msystems@asmusa.org.

Sincerely,

Rachel Mackelprang
Editor, mSystems

Journals Department
E-mail: mSystems@asmusa.org